



# Non-classical disproportionation revealed by photo-CIDNP NMR

Jakob Wörner[1], Jing Chen[1], Adelbert Bacher[2], and Stefan Weber[1]

[1]Institute of Physical Chemistry, Albert-Ludwigs-Universität Freiburg, Freiburg, 79104, Germany

[2]Department of Chemistry, Technical University of Munich, Garching, 85748, Germany

*Correspondence to*: Stefan Weber (Stefan.Weber@physchem.uni-freiburg.de)

**Abstract.** Photo-chemically induced dynamic nuclear polarization (photo-CIDNP) was used to observe the light-induced disproportionation reaction of 6,7,8-trimethyllumazine starting out from its triplet state to generate a pair of radicals comprising a one-electron-reduced and a one-electron oxidized species. Our evidence is based on the measurement of two marker proton hyperfine couplings, $A_{iso}(H(6\alpha))$ and $A_{iso}(H(8\alpha))$, which we correlated to predictions from density functional theory. The ratio of these two hyperfine couplings is reversed in the oxidized and the reduced radical species. Observation of the dismutation reaction is facilitated by the exceptional C–H acidity of the methyl group at position 7 of 6,7,8-trimethyllumazine and the slow proton exchange associated with it, which leads to NMR-distinguishable anionic (TML⁻) and neutral (TMLH) protonation forms.

## 1 Introduction

The first intentional synthesis of pteridine-2,4(1*H*,3*H*)-dione (**1**, see Scheme 1) was described by Kühling in 1894 (Kühling, 1894, 1895). Due to its blueish-green fluorescence even in very dilute aqueous solutions, the compound was named lumazine (Kuhn and Cook, 1937). More than 70 years after its first synthesis, **1** was recognized as a natural product and isolated from male ants of the species *Formica polyctena* (Schmidt and Viscontini, 1967) and more recently also from the leaves of *Brassica juncea* L. (brown mustard) (Sharma et al., 2018). Unsubstituted lumazine (**1**) is not widely distributed in Nature (Daniels et al., 2019); related compounds substituted at positions 6, 7 and 8 are of greater biological significance. Importantly, 6,7-dimethyl-8-ribityllumazine (**2**) is the direct biosynthetic precursor of riboflavin (**4**; vitamin B₂) (Masuda, 1957; Maley and Plaut, 1959; Plaut, 1960, 1963) whose derivatives flavin mononucleotide (FMN) and flavin adenine dinucleotide (FAD) are universally distributed and are involved in an amazing variety of essential biological processes; for recent reviews see (Walsh and Wencewicz, 2013; Piano et al., 2017). The biosynthetic precursor **2** affords riboflavin by a mechanistically unique dismutation that is catalyzed by the enzyme riboflavin synthase and does not require any cosubstrates or cofactors (Plaut and Harvey, 1971). Even more surprisingly, the dismutation affording **4** from **2** can also proceed non-enzymatically in neutral or acidic aqueous solution under an inert gas atmosphere (Rowan and Wood, 1963; Kis et al., 2001).



**Scheme 1**. Lumazine (**1**), 6,7-dimethyl-8-ribityllumazine (**2**), 6,7,8-trimethyllumazine (**3**) and riboflavin (**4**)

6,7-Dimethyl-8-ribityllumazine (**2**) acts as cofactor of lumazine protein (LumP), an optical transponder involved in bioluminescence of certain marine bacteria (Koka and Lee, 1979; Small et al., 1980). More recently, **2** has also been found in two members of the photolyase/cryptochrome protein family, namely cryptochrome B (CryB) from *Rhodobacter sphaeroides* (Geisselbrecht et al., 2012) and the photolyase-related protein B (PhrB) from *Agrobacterium tumefaciens* (Zhang et al., 2013). Both proteins belong to the new subclass of FeS bacterial cryptochromes and photolyases (BCP), also called CryPro. *R. sphaeroides* CryB controls light-dependent and singlet-oxygen-dependent gene expression of the photosynthetic apparatus (Frühwirth et al., 2012). Similar to *A. tumefaciens* PhrB, CryB has also repair activity for (6–4) photoproducts in photodamaged DNA (von Zadow et al., 2016). It has been speculated that **2** acts as antenna chromophore in this protein class (Geisselbrecht et al., 2012), however, it absorbs at shorter wavelengths than the essential FAD cofactor, which is the origin of light-induced one-electron transfer that initiates radical-pair spin chemistry in photolyases and cryptochromes (Biskup et al., 2009; Sheppard et al., 2017). Clearly, the precise role of **2** in this photolyase/cryptochrome subclade needs to be evaluated, in particular in its interplay with the FAD cofactor.

6,7-Dimethyl-8-ribityllumazine and certain structural analogs, e. g. 6,7,8-trimethyllumazine (**3**), exhibit anomalously high C–H acidity of the methyl group at position 7. For **2** and **3**, p$K_a$ values of 8.3 (Pfleiderer et al., 1966; Bown et al., 1986) resp. 9.9 (Pfleiderer et al., 1966; McAndless and Stewart, 1970; Bown et al., 1986) have been reported. Using [1]H and [13]C NMR, compound **3** has been found to form an anionic species under alkaline conditions which has been assigned a 7α-exomethylene motif. Compound **2** forms additionally several tricyclic ether anion species under the participation of the OH groups of the ribityl side chain attached at position 8 (Bown et al., 1986). Interestingly, riboflavin synthase selectively binds the 7α-exomethylene anion of **2**, which is believed to be crucial for the dismutation of **2** affording a stoichiometric mixture of riboflavin (**4**) and 5-amino-6-ribitylaninouracil. This reaction has been shown to proceed via a pentacyclic intermediate which was isolated using an inactive mutant of riboflavin synthase (Illarionov et al., 2001). Various pathways have been proposed for the riboflavin synthesis from **2** (Truffault et al., 2001; Gerhardt et al., 2002; Kim et al., 2010). For the non-enzymatic reaction, a quantum mechanical simulation favors a nucleophilic addition mechanism, which was calculated as the lowest energy pathway yielding riboflavin (Breugst et al., 2013).

In this contribution, we report on a process between the neutral and the anionic 6,7,8-substituted lumazine species **3**. Studies along these lines may ultimately shed light on the role of the related compound **2** in light-induced redox reactions of proteins from the CryPro subclade of photolyases and cryptochromes.



## 2 Experimental part

### 2.1 Sample preparation

6,7,8-Trimethyllumazine was prepared using a procedure described previously (Masuda, 1957) and purified using HPLC. For details, see Supplementary Material.

### 2.2 NMR and photo-CIDNP spectroscopy

NMR and photo-CIDNP experiments were performed as described previously (Pompe et al., 2019), using a Bruker Avance III HD NMR spectrometer (Bruker BioSpin GmbH, Rheinstetten, Germany) operating at 14.1 T and a proton resonance frequency of 600 MHz. Light excitation was achieved by coupling the output of a nanosecond-pulsed laser system, comprising an Nd:YAG laser source (Surelite I, Continuum, Santa Clara, CA, USA) in combination with a broadband optical parametric oscillator (OPO) (Continuum OPO PLUS), into an optical fibre with a diameter of 1 mm (Thorlabs, Dachau, Germany). The optical fibre was inserted into the NMR tube via a coaxial insert (Wilmad WGS-5BL). Photo-CIDNP difference spectra were recorded directly by using a pre-saturation pulse train to destroy thermal polarization prior to the laser flash (Goez et al., 2005). This avoids errors involved with the subtraction of light and dark spectra from separate experiments. A destructive phase cycle was additionally applied in which every second scan contained light excitation to avoid residual thermal NMR signals, especially contributions from the solvent peak (HDO) at 4.8 ppm.

### 2.3 Quantum chemical calculations

Molecular structures were drawn in "Avogadro 1.2.0" (Hanwell et al., 2012) and were subsequently pre-optimized using the MMFF94 force field (Halgren, 1996b, a). Geometry optimizations were performed in "Orca 4.0.1.2" (Neese, 2012, 2018) using the B3LYP functional (Stephens et al., 1994) and a TZVP basis set (Schäfer et al., 1994). The CPCM model (Barone and Cossi, 1998) was used to simulate water solvation. Calculations of hyperfine coupling constants and **g** matrices were then carried out by using a B3LYP functional together with the EPR-II basis set (Barone, 1996).

## 3 Results and Discussions

Deprotonation of 6,7,8-trimethyllumazine affords a structurally unusual 7-exomethylene anion (Beach and Plaut, 1970; Bown et al., 1986) subsequently designated TML$^-$, see Fig. 1. With a reported p$K_a$ around 9.9 (Pfleiderer et al., 1966), 6,7,8-trimethyllumazine exhibits extraordinarily strong C–H acidity. For photo-CIDNP experiments, samples with a predetermined ratio of 6,7,8-trimethyllumazine (the neutral molecule form is subsequently designated TMLH, see Fig. 1) and the cognate anion designated TML$^-$ were prepared by the addition of NaOD to a 6,7,8-trimethyllumazine solution (1–4 mM range) in 99 % D$_2$O. The TMLH:TML$^-$ ratio was monitored using $^1$H NMR at 600 MHz. In 6,7,8-trimethyllumazine, the three methyl





groups at positions 6, 7 and 8 and the hydrogen at N(3) afford $^1$H resonances. In D$_2$O, H(3) and the 7-methyl group exchange

protons with the bulk solvent (Beach and Plaut, 1970); the latter exchange follows first-order kinetics on a double-digit

minutes timescale (McAndless and Stewart, 1970). For the neutral TMLH species, we observed resonances of equal intensity

at chemical shift values of 3.91 ppm and 2.52 ppm, which we assigned as H(8$\alpha$) and H(6$\alpha$), respectively, based on reported

values in the literature (4.02 ppm, H(8$\alpha$); 2.61 ppm, H(6$\alpha$)) (McAndless and Stewart, 1970), see NMR spectra in Fig. 2. For

TML$^-$ in alkaline solution, chemical shift values have been reported for H(8$\alpha$) (3.15 ppm) and H(6$\alpha$) (2.10 ppm) (Beach and

Plaut, 1970). Accordingly, we assigned the resonances at 3.09 ppm and 2.06 ppm to H(8$\alpha$) and H(6$\alpha$), respectively. Based

on the integrals of the resonance lines of the two species, photo-CIDNP experiments were conducted at TML$^-$-to-TMLH

ratios of 1:11, 1:2, 1:1, 1.5:1, 7:1, and 10:1 at various pH values below, near or above the p$K_a$.


**Figure 1: CH acidity of the methyl group attached to position 7 in 6,7,8-trimethyllumazine. The deprotonated (anionic) form TML$^-$ can be drawn in various mesomeric structures, two of which are depicted on the right-hand side.**





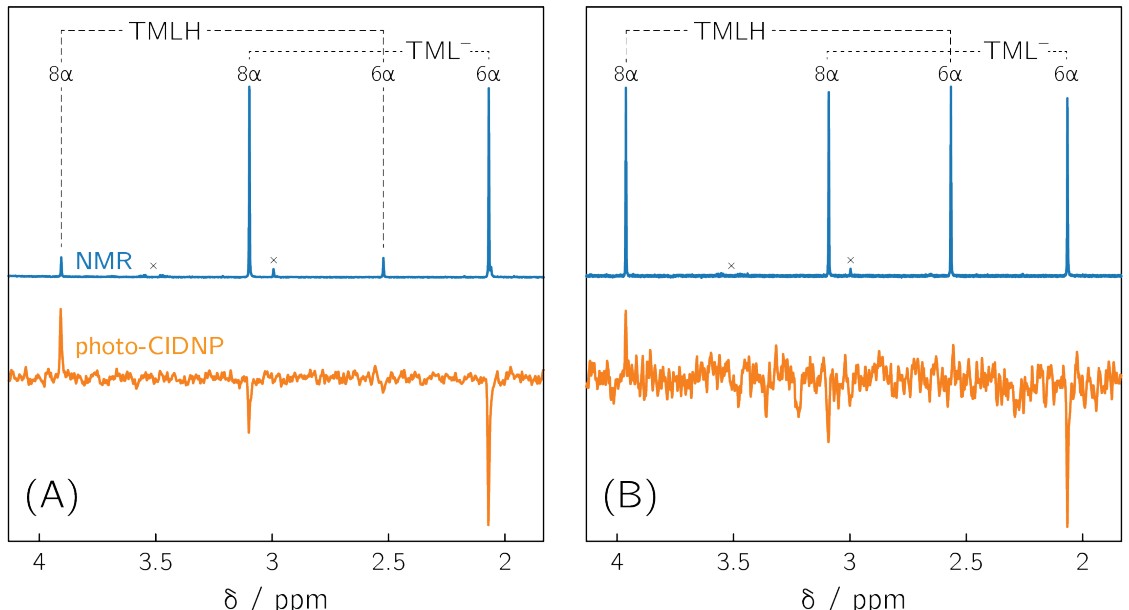

**Figure 2: ¹H NMR (blue curves) and photo-CIDNP data (orange curves) of alkaline solutions of 6,7,8-trimethyllumazine: (A) TMLH:TML⁻ ratio of 1:10; 6,7,8-trimethyllumazine concentration: 4 mM; excitation wavelength: 425 nm; (B) TMLH:TML⁻ ratio of 1:1; 6,7,8-trimethyllumazine concentration: 4 mM; excitation wavelength: 470 nm.**


Photoexcitation of the TMLH/TML⁻ solutions for ¹H photo-CIDNP experiments was performed using 6-ns pulses of an Nd:YAG-laser-pumped OPO adjusted to 425 or 470 nm (pulse energies of 8 mJ at 425 nm and 30 mJ at 470 nm). TMLH absorbs preferentially at these wavelengths because the long-wavelength absorbance of TML⁻ (364 nm) is blue-shifted with

respect to that of TMLH (402 nm) (Pfleiderer et al., 1966), see Fig. 3. Selected photo-CIDNP data are shown in Fig. 2. Three resonances exhibit substantial nuclear spin polarization: the position 6 and 8 methyl groups of signals of TML⁻, and the position 8 methyl group of TMLH. The protons of the position 6 methyl group of TMLH do not show appreciable hyperpolarization.

The occurrence of nuclear hyperpolarization upon photoexcitation of alkaline 6,7,8-trimethyllumazine samples provides

clear evidence for a photochemical reaction involving radical pair intermediates. Since 6,7,8-trimethyllumazine is the only organic species present that absorbs light in the visible range, we suggest disproportionation to take place under the given conditions, see Fig. 4: Light-initiated electron transfer from the anionic TML⁻ to the neutral TMLH generates the neutral radical TMLox• and initially the anionic radical TMLHred•⁻ as short-lived products. In the dark, backward electron transfer takes place to regenerate the initial species TML⁻ and TMLH, respectively.




**Figure 3: UV/vis spectra of TMLH (orange curve) and TML⁻ (blue curve), recorded in water and in 1M NaOH, respectively.**

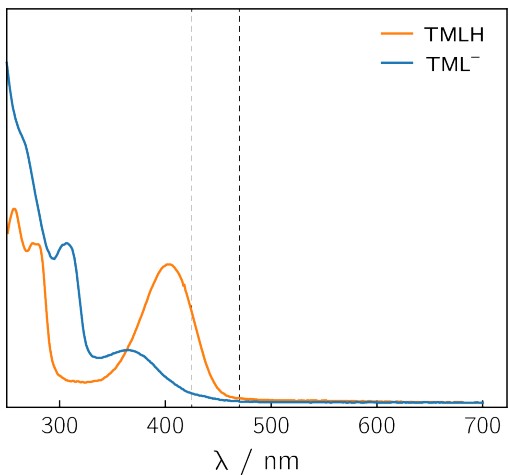


**Figure 4: Photoinduced reversible disproportionation of 6,7,8-trimethyllumazine.**

To corroborate this notion, we analysed intensities and signs of the hyperpolarized NMR resonances. In the high-field approximation, the applied time-resolved photo-CIDNP scheme using a pulsed laser as light source (Goez et al., 2005; Kuhn, 2013) renders signal intensities that are proportional to the isotropic hyperfine coupling constant $A_{\mathrm{iso}}$ of the respective



nucleus. In the present case with only two weakly coupled $^1$H spins per species, the relative enhancement factors can, in principle, be extracted by signal integration. Kaptein introduced a simple rule for the net polarization $\Gamma_i$ of a hyperpolarized resonance (Kaptein, 1971). $\Gamma_i$ results from the product of four signs and yields either "+" or "–" for an absorptive or an emissive signal, respectively:

$$\Gamma_i = \mu \times \varepsilon \times \mathrm{sgn}(\Delta g) \times \mathrm{sgn}(A_{\mathrm{iso},i}) \ . \tag{1}$$

The parameter $\mu$ is either "+" in case the radical pair is formed from a triplet precursor, or "–" in case of a singlet precursor. The reaction route following the formation of the intermediate radical pair determines the sign of $\varepsilon$, either "+" for recombination or "–" for dissociation, the latter leading to so-called escape products. The sign of the difference of the two (isotropic) $g$-factors of the two involved radicals, $\Delta g = g_1 - g_2$, depends on which of the two radical moieties comprising the radical pair is observed (see below). Finally, the sign of the isotropic hyperfine coupling constant ($A_{\mathrm{iso},i}$) of the respective

nucleus $i$ is of relevance.

   To apply Kaptein's sign rule for rationalizing the polarization of a particular resonance in the photo-CIDNP spectrum, we consider the following aspects: (i) Since the time delay between pulsed laser excitation and the radio-frequency pulse applied for the detection of the free-induction decay was chosen rather short (~80 ns), the detection of recombination products is supposed to be more likely than that of escape products, thus $\varepsilon \rightarrow$ "+". (ii) Little is known on the $g$-factors of

paramagnetic lumazine species. Early EPR studies were focussed on analyses of hyperfine patterns in EPR spectra from radicals of 6,7,8-trimethyllumazine (Ehrenberg et al., 1970) and derivatives thereof (Westerling et al., 1977), but did not report on their $g$-values. A more recent high-field EPR study considered 6,7-dimethyl-8-ribityllumazine bound to lumazine protein (Paulus et al., 2014). This cofactor was photo-reduced to yield the neutral 6,7-dimethyl-8-ribityllumazine radical, from which the isotropic $g$-factor of $2.0032 \pm 0.0001$ was obtained by averaging the principal values of the **g**-matrix. The

lack of respective experimental data for the specific 6,7,8-trimethyllumazine radicals involved in disproportionation, see Fig. 4, prompted us to perform quantum-chemical computations at the DFT level to calculate the necessary values. Calculated $g_{\mathrm{iso}}$ values were 2.0031 for the neutral radical TML$^{\mathrm{ox}\bullet}$ and 2.0034 for the anionic radical TMLH$^{\mathrm{red}\bullet-}$. Tentatively, the $g_{\mathrm{iso}}$-value of TMLH$^{\mathrm{red}\bullet-}$ may be compared to the measured value of protein-bound 6,7-dimethyl-8-ribityllumazine radical because both species result from one-electron reduction of their aromatic moiety; nevertheless, the substituents bound to the 8-position

and the protonation states of both radicals are different. Neglecting the unequal substitution at position 8, the neutral 6,7-dimethyl-8-ribityllumazine radical may be considered as the species obtained by protonation of TMLH$^{\mathrm{red}\bullet-}$ to yield TMLH$_2^{\mathrm{red}\bullet}$, see Fig. 4. Extrapolated to the realm of the related flavins that share similar hydrogen bonding motifs with the respective lumazines, see Scheme 1, the couple TMLH$^{\mathrm{red}\bullet-}$/ TMLH$_2^{\mathrm{red}\bullet}$ is expected to behave like anionic (Fl$^{\bullet-}$) / neutral (FlH$^\bullet$) flavin semiquinone radicals. For the latter, slightly larger $g_{\mathrm{iso}}$ values were observed for anion radicals than for neutral

radicals (Schleicher and Weber, 2012): ~2.0035 (Barquera et al., 2003; Okafuji et al., 2008) versus ~2.0034 (Fuchs et al.,



2002; Barquera et al., 2003; Schnegg et al., 2006), respectively, but the difference is quite small. This seems to also hold for the one-electron-reduced lumazine radicals: $g_{iso}$(TMLH$^{red\bullet-}$) > $g_{iso}$(neutral 6,7-dimethyl-8-ribityllumazine radical). (iii) Absolute values of most proton hyperfine couplings have been determined for a cationic (one-electron reduced) 6,7,8-trimethyllumazine radical species (Ehrenberg et al., 1970) and for the neutral 6,7-dimethyl-8-ribityllumazine radical (Paulus

et al., 2014). However, the signs of the $A_{iso}$ values have not been determined experimentally. Since the available hyperfine data from the literature are only of limited value for the interpretation of our photo-CIDNP spectra, we performed quantum-chemical computations also of the hyperfine structure of the 6,7,8-trimethyllumazine radicals under discussion. The $^1$H hyperfine couplings relevant for the interpretation of the photo-CIDNP NMR data are compiled in Table 1, together with the respective $g_{iso}$ values. A full set of hyperfine couplings from all other protons as well as from $^{13}$C and $^{15}$N nuclei can be

found in the Supplementary Material. Additionally, the respective data for two further one-electron reduced species have been included that potentially result from protonation of the anionic TMLH$^{red\bullet-}$ at N(1) or N(5). Importantly, for all one-electron reduced 6,7,8-trimethyllumazine radicals, DFT predicts for all one-electron reduced TML radicals isotropic hyperfine couplings of the H(8α) protons that are much larger than those of H(6α): $A_{iso}$(H(8α)) >> $A_{iso}$(H(6α)), see Table 1; EPR data are consistent with this finding (Ehrenberg et al., 1970). For the TML$^{ox\bullet}$ radical that results from TML$^-$ by

withdrawal of one electron, DFT predicts $A_{iso}$(H(6α)) > $A_{iso}$(H(8α)).

**Table 2: Isotropic g-values and selected isotropic methyl proton hyperfine couplings for various oxidized and reduced TML radicals.**

|  |  | TMLH$^{red\bullet-}$ |  | TMLH$_2$$^{red\bullet}$(H(5)) |  | TMLH$_2$$^{red\bullet}$(H(1)) |  | TML$^{ox\bullet}$ |  |
|---|---|---|---|---|---|---|---|---|---|
| $g_{iso}$ (DFT) |  | 2.0034 |  | 2.0033 |  | 2.0034 |  | 2.0031 |  |
| Proton |  | 6α | 8α | 6α | 8α | 6α | 8α | 6α | 8α |
| $A_{iso}$ (DFT) | abs. / MHz | −5.31 | +15.21 | +1.57 | +18.90 | −5.38 | +14.41 | +14.93 | +6.42 |
|  | rel. | −0.349 | 1 | 0.083 | 1 | −0.373 | 1 | 1 | 0.430 |
| $A_{iso}$ (CIDNP) | rel. | 0 | 1 | 0 | 1 | 0 | 1 | 1 | 0.439 |

Photo-excitation of an alkaline 6,7,8-trimethyllumazine solution (4 mM; TMLH:TML$^-$ ratio of 1:10) with 425-nm laser pulses resulted in a photo-CIDNP spectrum with both TML$^-$ resonances in emission, see Fig. 2(A). The ratio of the integrals of the signals assigned to H(6α) and H(8α) is 1:0.439. The situation is different for the signals assigned to TMLH: whereas the H(8α) resonance exhibits enhanced absorption, the one of H(6α) does not show significant hyperpolarization. Virtually the same polarization pattern is observed for a less alkaline 6,7,8-trimethyllumazine solution (4 mM) with a TMLH:TML$^-$

ratio of 1:1, i. e. pH ≈ p$K_a$, see Fig. 2(B). However, to obtain a discernible photo-CIDNP spectrum in this case, the excitation wavelength of our laser system had to be tuned to 470 nm. The higher wavelength was useful because of the rather high



TMLH concentration and the high absorbance associated therewith, which did not allow for sufficient photo-excitation of the active sample volume given the available output power of our laser source at 425 nm. Nevertheless, the obtained signal-to-noise ratio remained rather low and it even decreased upon further decreasing the amount of $TML^-$ relative to that of TMLH.

Observation of the photo-CIDNP effect at 470 nm is clear evidence for photo-excitation of TMLH rather than $TML^-$. The latter has very low absorbance at 425 nm and does virtually not absorb at 470 nm (Pfleiderer et al., 1966), see Fig. 3.

By far the highest signal-to-noise ratio of the photo-CIDNP data was obtained in the alkaline range at about one pH unit above the $pK_a$ of $TMLH/TML^-$. We could not conduct NMR experiments under more basic pH conditions because the high ion strength of our sample precluded proper tuning of the probehead. Additionally, we varied the 6,7,8-trimethyllumazine

concentration in a range between 1.0 and 4.0 mM. In all cases, photo-CIDNP revealed a hyperpolarization pattern similar to the one shown in Fig. 2(A).

To rationalize our findings, we correlated the relative intensities of the hyperpolarized NMR resonances obtained by photo-CIDNP to DFT predictions of hyperfine couplings of the various paramagnetic one-electron oxidized or reduced lumazine species involved in the suggested disproportionation scheme, see Table 1, Fig. 4 and Supplementary Material.

Upon backward electron transfer in the dark, i. e. radical pair recombination ($\varepsilon$ = "+"), hyperpolarization generated on the intermediate oxidized species $TML^{ox\bullet}$ and the reduced species $TMLH^{red\bullet-}$ (or a protonated neutral species $TMLH_2^{red\bullet}$ thereof) are transferred to the diamagnetic products $TML^-$ and TMLH, respectively.

Considering the hyperpolarized NMR resonances of $TML^-$ leads to $\Delta g = g(TML^{ox\bullet}) - g(TMLH^{red\bullet-}) < 0$ (see Table 1), i. e. the sign entering Eq. (1) becomes negative: $sgn(\Delta g)$ = "–". Furthermore, the hyperfine couplings from the precursor state

$TML^{ox\bullet}$ are relevant. The calculated isotropic H(6$\alpha$) and H(8$\alpha$) hyperfine couplings of $TML^{ox\bullet}$ are positive, hence $sgn(A_{iso,i})$ = "+" for $i \in \{H(6\alpha), H(8\alpha)\}$. Given the fact that both resonances are emissively polarized, i. e. $\Gamma_i$ = "–", requires $\mu$ = "+" for the precursor multiplicity. Consequently, radical pair formation must proceed from a triplet-state precursor of radical pair formation, i. e. from $^3$TMLH, which is generated by intersystem crossing from an excited singlet state ($^{1*}$TMLH) of TMLH. Plotting the photo-CIDNP intensities with respect to the hyperfine couplings obtained using DFT (see Fig. 5) reveals nearly

perfect correlation: a linear regression fit constrained to go through the origin yields a slope of 14.88 MHz and $R^2 \approx 1$, see Fig. 5(B). Observation of hyperpolarized resonances of H(6$\alpha$) and H(8$\alpha$) from $TML^-$ both in emission and in an intensity ratio that correlates well with hyperfine coupling computations from DFT provides clear evidence for the existence of the oxidized TML species $TML^{ox\bullet}$, which is a redox state of lumazine that up to now had not been substantiated by experiments.

If we retain the signs of $\varepsilon$ ("+", recombination) and $\mu$ ("+", triplet precursor), and reverse the sign of $\Delta g$ to "+" (because

$\Delta g = g(TML^{red\bullet-}) - g(TML^{ox\bullet}) > 0$), then we expect for the H(8$\alpha$) resonance of TMLH because of $\Gamma_i$ = "+" (absorptive resonance) a positive isotropic hyperfine coupling constant of its paramagnetic precursor state. A very small hyperfine coupling near or equal to zero is expected for H(6$\alpha$) as hardly any hyperpolarization is observed for these nuclei in the photo-CIDNP spectrum. This situation is reversed as compared to that of $TML^-$, for which the H(6$\alpha$) resonance experiences





much stronger hyperpolarization than H(8α). Our DFT calculations confirm a large and positive value for $A_{iso}$(H(8α)) of
TMLH$^{red•-}$ but also predict a negative hyperfine coupling of substantial absolute value for $A_{iso}$(H(6α)). This latter finding is
clearly not supported by the photo-CIDNP data shown in Fig. 2. Therefore, we have extended our DFT studies of 6,7,8-
trimethyllumazine radicals to protonated variants of TMLH$^{red•-}$, namely the neutral species TMLH$_2$$^{red•}$(H(1)) (protonated at
N(1)) and TMLH$_2$$^{red•}$(H(5)) (protonated at N(5)), see Table 1.

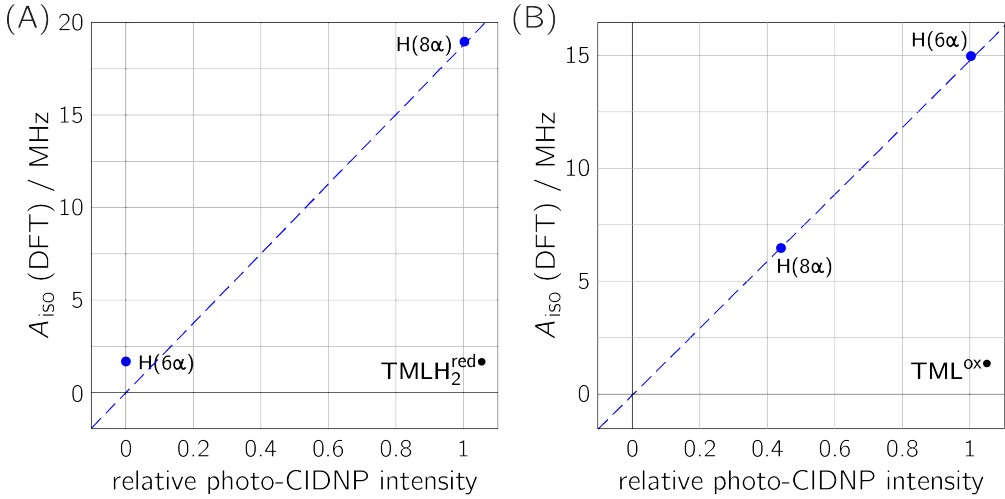


**Figure 5: Correlation of relative photo-CIDNP intensities to predicted $A_{iso}$ values from DFT, see Table 1: (A) TMLH$_2$$^{red•}$ protonated at N(5), and (B) TML$^{ox•}$.**

Protonation of TMLH$^{red•-}$ at N(1) to yield TMLH$_2$$^{red•}$(H(1)) does not significantly alter the isotropic hyperfine couplings
of H(6α) and H(8α); also $g_{iso}$ is virtually unaffected, see Table 1. However, addition of a proton at N(5) to yield
TMLH$_2$$^{red•}$(H(5)) does shift both hyperfine couplings to more positive values. $A_{iso}$(H(6α)) even changes its sign and assumes
a small positive value, but more than 10 times smaller than that of $A_{iso}$(H(8α)). Our photo-CIDNP data thus support rapid
protonation of TMLH$^{red•-}$ to yield the neutral radical species TMLH$_2$$^{red•}$(H(5)). Correlation of the relative photo-CIDNP
intensities of the TMLH proton resonances (the ratio of the integrals of the signals assigned to H(8α) and H(6α) is 1:0) to
DFT predictions of the respective $A_{iso}$ values yields the plot shown in Fig. 5(A), from which the slope of 18.9 MHz is
extracted ($R^2$ = 0.984).

The slopes of the straight lines through the origin in the correlation plots of TML$^-$ (Fig. 5(B)) and TMLH (Fig. 5(A)) are
clearly different: 14.8 MHz versus 18.9 MHz, respectively. Furthermore, even though $A_{iso}$ of H(8α) in TMLH$_2$$^{red•}$ has a
larger value than $A_{iso}$ of H(6α) in TML$^{ox•}$, the corresponding photo-CIDNP intensity of the resonance in the recombination
product is significantly smaller than that of the most intense signal of the respective counter radical. This may have several
reasons: (i) Introduction of an additional proton at N(5) introduces a further large hyperfine coupling ($A_{iso}$(H(5)) of



substantial anisotropy, see Supplementary Material. This could enhance relaxation by which hyperpolarized spin-state population decays to the population at thermal equilibrium. (ii) Hyperpolarization could also be dissipated into the solvent upon radical-pair recombination. Backward electron transfer from TMLH$_2$$^{red•}$ yields the diamagnetic TMLH$_2$$^+$, a species that will certainly deprotonate quickly to regenerate TMLH, especially given the alkaline conditions. Hence, intermediate

electron-spin redistribution leading to build-up of hyperpolarization at H(5) will likely be transferred to the surroundings on release of this proton. (iii) Despite the fact that electron exchange has been observed in other systems leading to a decay of hyperpolarization (Closs and Sitzmann, 1981), we consider such a mechanism along the scheme *TMLH$^{red•-}$ + TMLH → *TMLH + TMLH$^{red•-}$ (the asterisks denote nuclear spin polarization) less likely given that at the elevated pH values under consideration neutral TMLH is present only at rather low concentrations.


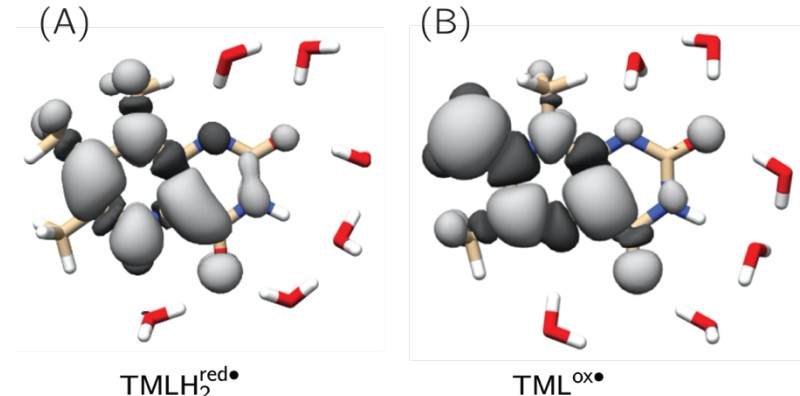

**Figure 6: SOMO of observed 6,7,8-trimethyllumazine radical species. Dark and light grey contours denote negative and positive signs of the molecular wavefunctions. (A) TMLH$_2$$^{red•}$ protonated at N(5); (B) TML$^{ox•}$.**

Figure 6 shows the singly-occupied molecular orbitals (SOMO) of the neutral radicals TMLH$_2$$^{red•}$(H(5)) and TML$^{ox•}$. Positive and negative signs of the frontier orbitals are depicted with light and dark grey shading, respectively. Mere inspection gives a hint for the quite different ratios of isotropic hyperfine couplings of H(8α) and H(6α) in both species: The amplitude of the SOMO at the 6-methyl group of TMLH$_2$$^{red•}$(H(5)) is small as compared to that of TML$^{ox•}$. For the 8-methyl group the opposite trend is observed. Considerable SOMO amplitudes are observed for H(5) in TMLH$_2$$^{red•}$(H(5)) which leads

to a strong and anisotropic hyperfine coupling of this exchangeable proton. This explains the dissipation of hyperpolarization into the solvent on backward electron transfer leading from TMLH$_2$$^{red•}$ to TMLH and H$^+$.

   The SOMOs of TMLH$_2$$^{red•}$(H(5)) and TML$^{ox•}$ significantly differ in terms of wavefunction amplitudes and signs, in particular at the respective π-system of the pyrazine ring. Very high amplitude is observed at C(7α) of TML$^{ox•}$. This gives a hint that the electronic structure of the related one-electron reduced TML$^-$ may be better represented by a C(7α)-carbanion

rather than a 7α-exomethylene motif at position 7, see Fig. 1. Clearly, the two observable proton resonances per radical



species and their hyperpolarization detected using photo-CIDNP NMR are insufficient to draw a precise picture of the delocalization of the unpaired electron spin density over the carbons and nitrogens in the heterocyclic cores. To learn more on the electron-spin distributions in the two radicals generated by disproportionation of **3** and to further corroborate the existence of the oxidized species TML$^{ox\bullet}$, we plan further photo-CIDNP experiments on specifically designed $^{13}$C and $^{15}$N

isotopologs of **2** and **3**.

## 4 Conclusions

Using photo-CIDNP NMR we have discovered a disproportionation reaction upon photoexcitation of alkaline solutions of 6,7,8-trimethyllumazine. In its classical definition, disproportionation refers to a "reversible or irreversible transition in which species with the same oxidation state combine to yield one of higher oxidation state and one of lower oxidation state"

(McNaught and Wilkinson, 1997). This includes redox reactions of the type: $2\,X \rightarrow X^{ox\bullet+} + X^{red\bullet-}$. From the perspective of oxidation states, this scheme applies to 6,7,8-trimethyllumazine, which upon photo-induced electron transfer generates a pair of radicals comprising a species devoid of one electron (TML$^{ox\bullet}$) and another with an excess electron (TMLH$^{red\bullet-}$). However, our proposed mechanism deviates from the classical disproportionation scheme in so far as (i) the reaction is initiated by light, and (ii) that a redox reaction takes place between two *different* protonation states of 6,7,8-trimethyllumazine, i. e.

between TMLH and TML$^-$. Clearly, disproportionation starts out from photoexcitation of TMLH (notably, only TMLH has significant absorption at 470 nm). Once the triplet state of TMLH is formed by intersystem crossing from an excited singlet state of this molecule, it abstracts an electron from TML$^-$, thereby generating a pair of interacting radicals, see Fig. 4. Interestingly, one quite unusual radical species is generated by this disproportionation that has not been reported before: TML$^{ox\bullet}$. The existence of a species in such a high redox state was speculated upon in triplet quenching of the unsubstituted

lumazine (**1**) (Denofrio et al., 2012); one should however keep in mind that the aromatic moiety of 6,7,8-trimethyllumazine (**3**) differs from that of the unsubstituted lumazine. A similar species was proposed for triplet quenching of the related flavins (Görner, 2007), but convincing experimental evidence on the existence of a species in such a high oxidation state was still lacking for both cases until now. By detecting two important hyperfine couplings, we provide strong evidence for the existence of 6,7,8-trimethyllumazine in a further high redox state. Clearly, we owe this success to two peculiarities of 6,7,8-

trimethyllumazine: (i) the extraordinary acidity of its 7-methyl group which compares to that of the ammonium ion, and (ii) its proton exchange on a time scale that is slow compared to that of NMR and which consequently leads to distinguishable anionic (TML$^-$) and neutral (TMLH) protonation forms in terms of NMR properties. Flavins by comparison do not exhibit a corresponding acidity of their methyl groups. Therefore, our photo-CIDNP detection scheme is not readily extendable to the realm of flavins for a proof of the existence of the speculative FAD$^{ox\bullet+}$ species.



Our data on 6,7,8-trimethyllumazine provide evidence for an extended range of redox states of lumazines in general. Further studies on lumazine-mediated photocatalysis will show whether the existence of species of the type of TML$^{ox\bullet}$ will be involved that shed new light on the role of 6,7-dimethyl-8-ribityllumazine as chromophore, e. g. in the new class of recently discovered CryB cryptochromes (Geisselbrecht et al., 2012) and PhrB photolyases (Oberpichler et al., 2011; Zhang et al., 2013) of the CryPro subclade. Further studies will be conducted on the suitability of TML as a photosensitizer.

## 5 Acknowledgements

SW thanks the SIBW/DFG for financing NMR instrumentation that is operated within the MagRes Center of the Albert-Ludwigs-Universität Freiburg. JC is thankful for a fellowship from the China Scholarship Council. We thank Ursula Friedrich for HPLC purification of 6,7,8-trimethyllumazine.

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
