# Peer review of "Non-classical disproportionation revealed by photo-CIDNP NMR"

_Magnetic Resonance, 2021_

## Referee Comment (RC2)

The manuscript "Non-classical disproportionation revealed by photo-CIDNP NMR" by Weber and co-authors describes the results obtained by photo-CIDNP in the disproportionation reaction of  compound 3, called 6,7,8-trimethyllumazine in its triplet state with 6,7,8-trimethyllumazine in the ground state. The authors applied the pulsed variant of CIDNP detection with multiple repetition of a short laser pulse followed by an RF pulse for NMR detection.

In 2011 direct proportionality between individual amplitudes of geminate CIDNP and the corresponding hyperfine coupling constants in a multinuclear radical pair at high magnetic field was theoretically predicted by the brilliant scientist Konstantin Ivanov, who too early passed away on 05.03 2021. The linearity was explained and verified in the publication of Morozova, O. B.; Ivanov, K. L.; Kiryutin, A. S.; Sagdeev, R. Z.; Köchling, T.; Vieth, H.-M.; Yurkovskaya, A. V., Time-resolved CIDNP: an NMR way to determine the EPR parameters of elusive radicals. *Phys. Chem. Chem. Phys.* **2011,** *13* (14), 6619-6627. Prior to that, the Adrian model (F. J. Adrian, *J. Chem. Phys.*, 1971, 54, 3918–3923) with a rather clumsy sum over all multiple nuclear spin projections was widely used for CIDNP simulation. Please, refer to these publications in the manuscript.

The results obtained are very interesting and the manuscript is written quite nicely. However, I agree with the comment of referee 1, that the general reaction scheme is needed to clarify the chemical processes in aqueous solution leading to CIDNP formation.
For the special issue dedicated to Prof. Robert Kaptein who remarkably contributed to CIDNP as a mature science, we decided to assist the authors of this nice manuscript and propose such a reaction scheme to show the process resulting in CIDNP formation. The scheme is shown below. In a reaction involving e-transfer coupled with proton transfer (PCET) between 6,7,8-trimethyllumazine in its triplet state and 6,7,8-trimethyllumazine in the ground state the primary triplet radical pair of (TMLH$^{red\bullet-}$ TML$^{ox\bullet}$ ) is formed. It can either recombine or reversibly form (TMLH$_2$$^{red\bullet}$ TML$^{\bullet\ ox}$) via protonation-deprotonation as shown in the scheme. On a second reaction path the radical pair disproportionates with formation of the same products (the arrow indicating the release of a proton (-H$^+$)), but with a different CIDNP pattern.

The resulting CIDNP pattern detected in the diamagnetic product of these reversible reactions is formed at least in two pairs. For each pair, the own proportionality relationship between HFCCs and CIDNP holds. For a similar situation, in the reaction of carboxy benzophenones, it was proposed to utilize a linear combination of proportionality relationships to find the share of several radical pairs that contribute to the CIDNP signal. (Morozova, O. B.; Panov, M. S.; Fishman, N. N.; Yurkovskaya, A. V., Electron transfer vs proton-coupled electron transfer as the mechanism of reaction between amino acids and triplet-excited benzophenones revealed by time-resolved CIDNP. *Phys. Chem. Chem. Phys*. 2018, 20 (32), 21127-21135). This procedure works very nicely for other cases as well (see examples in the recent paper "Molecular features toward high photo-CIDNP hyperpolariztion explored through the oxidocyclization of tryptophan" **Phys. Chem. Chem. Phys.**, 2021, Advance Article https://doi.org/10.1039/D0CP06068B).

When applied to the reaction of the methylated lumazine the result is shown in figure 1 by the red line and red symbols. Indeed, the linear combination of two radical pairs with shares of 0.23 and 0.77 provides perfect linearity with $R^2=1$.

Moreover, the correct approach to proportionality of CIDNP vs HFCCs is to use one common normalization of CIDNP intensities for the whole CIDNP spectrum, but not different ones for individual patterns from different protonation forms as it is done in Table 2, and to combine proportionalities in a single plot (see Fig. 1). In this case, different slopes (that did not receive an explanation by the authors) are clearly seen making it easier to follow the corresponding discussion.

[Figure]

Figure 1. Correlation between calculated HFCCs of TML radicals, and CIDNP intensities of corresponding protons of TMLH and TML⁻, detected during irradiation of TML solution. CIDNP intensities are multiplied by the corresponding sgn(Δ).

Such a way of presenting the experimental data makes the interpretation of CIDNP data very straightforward. Since the interconversion case by protonation and deprotonation for two radical pairs is revealed and explained in full accordance with the *classical* CIDNP theory proposed by Robert Kaptein, I strongly recommend to remove the word "non-classical" from the title or rephrase it. The CIDNP application to the studied reactions once again demonstrates the predictive power of CIDNP introduced by Prof. Robert Kaptein whose 80[th] anniversary we all celebrate!

The paper can be accepted after a revision as described above.

---

## Author Comment (AC1)

**Author comment for „mr-2021-22"**
**"Non-classical disproportionation revealed by photo-CIDNP NMR"**

We thank the two reviewers for their valuable comments, questions and suggestions that will help improve our contribution to the Robert Kaptein Festschrift.

**mr-2021-22-rc1:**

"Research on photolyases and cryptochromes is presently a hot field. Cryptochromes are even considered to be involved in "animal detected magnetic resonance". Therefore, an update on flavin-related compounds, their capacity to form radical pairs and to build up hyperpolarization is highly desirable. The manuscript revealed the existence of a transient high redox state of 6,7,8-trimethyllumazine by means of photo-CIDNP NMR. The compound is able to form upon illumination radical pairs with its two different protonation states. The experimental isotropic hyperfine coupling constants are obtained and correlate well with DFT data. Combining DFT and CIDNP, the careful analysis of the data on the basis of Kaptein's sign rule is convincing and allows a remarkable reconstruction of a reaction dynamics involving photochemistry, spin-chemistry as well as proton transfer."

We appreciate that this reviewer considers our contribution topical and the analysis well carried out. Thank you!

"As for the title, I feel that the authors are unnecessarily opening a discussion that is not really helpful. Why should a disproportionation ($2A = A^+ + A^-$) not also be able to be triggered photochemically ($A^* + A = A^+ + A^-$)? In the textbook by Klessinger and Michel, photochemically triggered disproportionations are indeed discussed (for example, as a result of Norrish I). One could also ask whether a system $A + AH$ really "disproportionates", since $A$ and $AH$ are different species. Nevertheless, it seems to me that a very special CIDNP-capable donor-acceptor system is present here, perhaps deserving the term "non-classical"."

In the suggested textbook by Klessinger and Michel we could not find any example of a disproportionation that takes place *directly* following light excitation and promotion into an excited multiplet state (excited singlet or triplet). The examples listed therein have disproportionation steps at later stages of the overall reaction and are typically irreversible. The reviewer is correct in stating that $A (= TML^-)$ and $AH (=TMLH)$ in 6,7,8-trimethyllumazine (TML) are different species, but they are protonation forms of the same molecular entity and in aqueous solution they are obligatorily coexistent. Whether there are other cases of photo-disproportionations of the type in 6,7,8-trimethyllumazine is very difficult (if not impossible) to find out. At least, there is no example we are aware of. Therefore, we consider the photoinduced disproportionation of 6,7,8-trimethyllumazine as "non-classical", but could also accept the terms "non-standard" or "atypical" instead.

This reviewer asks two questions:

"From the Figure 6B, it seems like the H(7α) also enjoys very high amplitude of electron spin density, but in the photo-CIDNP spectrum, there is no polarization at H(7α). How to understand such a contradiction?"

The methyl group at position 7 is acidic with a p$K$ value not far from that of the ammonium ion – which is quite unusual. In $D_2O$ all three protons are replaced with deuterons and consequently there are no proton resonances observed both in NMR and in photo-CIDNP, see line 89–91 of the original manuscript. We will emphasize this proton-to-deuteron exchange more explicitly in a revised version.

"In the supplementary information, Table S2, Table S3, there are three times H(6α), three times H(7α), and three times H(8α). What do they refer to?"

These are the individual hyperfine couplings of the three methyl protons per methyl group in the rigid molecule. Their isotropic hyperfine values are averaged (due to fast methyl rotation on the NMR timescale in aqueous solution and at room temperature) to yield the isotropic hyperfine values listed in Table 1 of the main text.

This reviewer has also minor comments:

"The "R." of "Rhodobacter sphaeroides" needs to be introduced (line 35)."

This will be corrected in a revised version. The same will be done with *Agrobacterium* (*A.*) *tumefaciens*.

""it absorbs at shorter wavelengths" (line 41): Should an antenna not exactly do that? Might be helpful to mention the absorption maxima."

This depends on the type of antenna and the wavelength range that shall be harvested. Why cryptochromes harbor 6,7-dimethyl-8-ribityllumazine as antenna, which has a blue-shifted absorption maximum ($\lambda_{max}$ = 420 nm) with respect to the flavin cofactor ($\lambda_{max}$ around 450 nm), cannot be explained straightforwardly in light of the fact that the flavin itself absorbs significantly near 420 nm and also shows an absorption maximum around 380 nm. Absorbance of an antenna could also be red-shifted as long as there is overlapping absorbance with the moiety eventually "receiving" the excitation energy.

The absorption maxima will be mentioned in the text in a revised version of the manuscript.

""operating at 14.1 T and a 1H freq of 600 MHz" (line 66): Sounds to be two different things."

This will be reworded in a revised version of the manuscript.

""fibre" (line 70) = fiber."

This will be corrected in a revised version of the manuscript.

"Fig Caption Fig 3: Dashed lines show laser excitation."

The caption of Fig. 3 will be supplemented accordingly.

"Line 212: radical pair formation from a triplet precursor: Is that surprising considering flavin photo-chemistry?"

One might expect a triplet precursor when comparing 6,7,8-trimethyllumazine to a flavin. However, 6,7,8-trimethyllumazine has one aromatic ring less than flavin. Hence, the expectation of a triplet pathway is in our opinion not obvious. Besides, there is at least one example of a flavin performing electron transfer from its excited singlet state: the BLUF (blue light using flavin) domain.

"Line 244: To what refers the "this"?"

This sentence will be reworded in a revised version of the manuscript.

"Line 253: The use of the asterisk in a text of photochemical relevance might be misleading."

We will choose a different symbol for indicating hyperpolarization.

**mr-2021-22-rc2:**

"The manuscript "Non-classical disproportionation revealed by photo-CIDNP NMR" by Weber and co-authors describes the results obtained by photo-CIDNP in the disproportionation reaction of the compound 3, called 6,7,8-trimethyllumazine in its triplet state with 6,7,8-trimethyllumazine in the ground state. The authors applied the pulsed variant of CIDNP detection with multiple repetition of a short laser pulse followed by an RF pulse for NMR detection.

In 2011 direct proportionality between individual amplitudes of geminate CIDNP and the corresponding hyperfine coupling constants in a multinuclear radical pair at high magnetic field was theoretically predicted by the brilliant scientist Konstantin Ivanov, who too early passed away on 05.03 2021. The linearity was explained and verified in the publication of Morozova, O. B.; Ivanov, K. L.; Kiryutin, A. S.; Sagdeev, R. Z.; Köchling, T.; Vieth, H.-M.; Yurkovskaya, A. V., Time-resolved CIDNP: an NMR way to determine the EPR parameters of elusive radicals. *Phys. Chem. Chem. Phys.* 2011, *13* (14), 6619–6627. Prior to that, the Adrian model (F. J. Adrian, *J. Chem. Phys.*, 1971, 54, 3918–3923) with a rather clumsy sum over all multiple nuclear spin projections was widely used for CIDNP simulation. Please, refer to these publications in the manuscript."

These two publications will of course be mentioned in a revised version of the manuscript:
"In the high-field approximation, the applied time-resolved photo-CIDNP scheme using a pulsed laser as light source (Closs et al., 1985; Goez 135 et al., 2005; Kuhn, 2013) renders signal intensities that are proportional to the isotropic hyperfine coupling constant $A_{iso}$ of the respective nucleus (Adrian, 1971; Morozova et al., 2011)." and "To rationalize our findings, we correlated the relative intensities of the hyperpolarized NMR resonances obtained by photo-CIDNP to DFT predictions of hyperfine couplings of the various paramagnetic one-electron oxidized or reduced lumazine species involved in the suggested disproportionation scheme along a procedure introduced by Ivanov and Yurkovskaya (Morozova et al., 2011), see Table 1, Fig. 5 and Supplementary Material."

"The results obtained are very interesting and the manuscript is written quite nicely. However, I agree with the comment of referee 1, that the general reaction scheme is needed to clarify the chemical processes in aqueous solution leading to CIDNP formation.
For the special issue dedicated to Prof. Robert Kaptein who remarkably contributed to CIDNP as a mature science, we decided to assist the authors of this nice manuscript and propose such a reaction scheme to show the process resulting in CIDNP formation. The scheme is shown below. In a reaction involving e-transfer coupled with proton transfer (PCET) between 6,7,8- trimethyllumazine in its triplet state and 6,7,8-trimethyllumazine in the ground state the primary triplet radical pair of $(TMLH^{red\bullet-} \ TML^{ox\bullet})$ is formed. It can either recombine or reversibly form $(TMLH_2^{red\bullet} \ TML^{ox\bullet})$ via protonation-deprotonation as shown in the scheme. On a second reaction path the radical pair disproportionates with formation of the same products (the arrow indicating the release of a proton (– H$^+$), but with a different CIDNP pattern."

We appreciate the suggestion by this reviewer and have modified the general reaction scheme accordingly, see new Figure 4:

**Figure 4:** Photoinduced reversible disproportionation of 6,7,8-trimethyllumazine. ISC, intersystem crossing.

"The resulting CIDNP pattern detected in the diamagnetic product of these reversible reactions is formed at least in two pairs. For each pair, the own proportionality relationship between HFCCs and CIDNP holds. For a similar situation, in the reaction of carboxy benzophenones, it was proposed to utilize a linear combination of proportionality relationships to find the share of several radical pairs that contribute to the CIDNP signal. (Morozova, O. B.; Panov, M. S.; Fishman, N. N.; Yurkovskaya, A. V., Electron transfer vs proton-coupled electron transfer as the mechanism of reaction between amino acids and triplet-excited benzophenones revealed by time-resolved CIDNP. *Phys. Chem. Chem. Phys.* 2018, 20 (32), 21127-21135). This procedure works very nicely for other cases as well (see examples in the recent paper "Molecular features toward high photo- CIDNP hyperpolariztion explored through the oxidocyclization of tryptophan" *Phys. Chem. Chem. Phys.*, 2021, Advance Article https://doi.org/10.1039/D0CP06068B)."

We will reference these two publications.

"When applied to the reaction of the methylated lumazine the result is shown in figure 1 by the red line and red symbols. Indeed, the linear combination of two radical pairs with shares of 0.23 and 0.77 provides perfect linearity with $R^2=1$."

When writing this manuscript, we were considering a linear combination of two radical pairs exactly as suggested by this reviewer. We did not dare to include such an analysis in the submitted version because hyperfine predictions by DFT (as used here for the correlation to photo-CIDNP intensities) are in many cases not "exact": Computed hyperfine couplings depend on the used functional, basis set and molecular geometry. Therefore, the weights of $TMLH_2^{red\bullet}(H(5))$ and $TMLH^{red\bullet-}$ should be treated with caution as the choice of other parameters for DFT computation might result in different hyperfine values. Nevertheless, the suggestion by this reviewer prompted us to be bolder so that a quantitative analysis based on two different radical pairs will be included. Figure 5 will be modified accordingly.

[Figure]

**Figure 5:** Correlations of relative photo-CIDNP intensities to predicted $A_{iso}$ values from DFT. $TML^{ox\bullet}$ (blue line and filled squares): slope = 0.0672 MHz$^{-1}$, $R^2$ = 0.9996; $TMLH^{red\bullet-}$ (orange dotted line and open circles): slope = 0.0277 MHz$^{-1}$, $R^2$ = 0.7827; $TMLH_2^{red\bullet}(H(5))$ (orange dashed line and filled circles): slope = 0.0248 MHz$^{-1}$, $R^2$ = 0.9863; linear combination of $TMLH^{red\bullet-}$ and $TMLH_2^{red\bullet}(H(5))$ with a ratio of 0.228:0.772 (orange drawn line and half-filled circles): slope = 0.0262 MHz$^{-1}$, $R^2$ = 1. CIDNP intensities were multiplied by the respective $sgn(\Delta g)$.

„Moreover, the correct approach to proportionality of CIDNP vs HFCCs is to use one common normalization of CIDNP intensities for the whole CIDNP spectrum, but not different ones for individual patterns from different protonation forms as it is done in Table 2, and to combine proportionalities in a single plot (see Fig. 1). In this case, different slopes (that did not receive an explanation by the authors) are clearly seen making it easier to follow the corresponding discussion."

We will changed Figure 5 accordingly, see above. Explanations of the different slopes were given in the original text, see lines 241 to 254.

„Such a way of presenting the experimental data makes the interpretation of CIDNP data very straightforward."

"Since the interconversion case by protonation and deprotonation for two radical pairs is revealed and explained in full accordance with the ***classical*** CIDNP theory proposed by Robert Kaptein, I strongly recommend to remove the word "non-classical" from the title or rephrase it."

The word "non-classical" does not refer to the applicable CIDNP theory but to the type of disproportionation as is manifested in 6,7,8-trimethyllumazine photochemistry. As mentioned in the response to reviewer 1 (mr-2021-22-rc1), we could also accept the terms "non-standard" or "atypical" instead of "non-classical".

„The CIDNP application to the studied reactions once again demonstrates the predictive power of CIDNP introduced by Prof. Robert Kaptein whose 80th anniversary we all celebrate!

The paper can be accepted after a revision as described above."

We thank also this reviewer for the many valuable suggestions that helped improve this contribution.